# Evaluating Approaches to Surrogate ODE-Based Modelling of Diffusion Bridges

**Maria Khilchuk, Vladimir Latypov, Pavel Kleshchev & Alexander Hvatov**
NSS Lab
ITMO University
Saint-Petesburg, 197101, Russia
{mdkhilchuk,donrumata,pskleshchev,alex_hvatov}@itmo.ru

## Abstract

Diffusion and Schrödinger Bridge models have established state-of-the-art performance in generative modeling but are often hampered by significant computational costs and complex training procedures. While continuous-time bridges promise faster sampling, overparameterized neural networks describe their optimal dynamics, and the underlying stochastic differential equations can be difficult to integrate efficiently. This work introduces a novel paradigm that uses surrogate models to create simpler, faster, and more flexible approximations of these dynamics. We propose two specific algorithms: SINDy Flow Matching (SINDy-FM), which leverages sparse regression to identify interpretable, symbolic differential equations from data, and a Neural-ODE reformulation of the Schrödinger Bridge (DSBM-NeuralODE) for flexible continuous-time parameterization. Our experiments on Gaussian transport tasks and MNIST latent translation demonstrate that these surrogates achieve competitive performance while offering dramatic improvements in efficiency and interpretability. The symbolic SINDy-FM models, in particular, reduce parameter counts by several orders of magnitude and enable near-instantaneous inference, paving the way for a new class of tractable and high-performing bridge models for practical deployment.

## 1 Introduction

Generative modeling has become the dominant paradigm in modern machine learning, with diffusion-based approaches establishing state-of-the-art performance in capturing complex data distributions. These models have succeeded in domains ranging from image synthesis Nichol and Dhariwal (2021); Dhariwal and Nichol (2021); Rombach et al. (2022); Chen et al. (2024); Lomurno et al. (2024); Liu et al. (2023) to molecular design Liu et al. (2022); Prakash et al. (2025); Duan et al. (2025); Atik Ahamed et al. (2025). However, this performance comes at a significant computational cost, primarily due to the slow, iterative nature of the sampling process Cao et al. (2025); Liu et al. (2025a); Issachar et al. (2025). The foundational strength of standard diffusion models lies in their well-defined, discrete Markov chain of forward-noising and reverse-denoising steps. This process progressively corrupts data with Gaussian noise, then learns a parameterized model to reverse the corruption, enabling sample generation from pure noise. While this paradigm is powerful and widely applicable, its sequential nature requires a large number of network evaluations to produce a single sample, which remains prohibitively slow for many real-world applications Cao et al. (2025); Xu et al. (2025); Chen and Deng (2025); Liu et al. (2025b).

The problem can be reformulated in continuous time using dynamic bridges, most notably the Schrödinger Bridge (SB) Shi et al. (2023); Liu et al. (2022); Gushchin et al. (2024). This framework seeks a continuous stochastic process that directly maps a simple initial distribution (e.g., noise) to a complex target data distribution, effectively constructing a bridge between them. By moving from discrete steps to a continuous flow, these methods promise a significant reduction in the number of function evaluations required for sampling. However, this conceptual elegance comes with trade-offs: the problem statement becomes more constrained, and training algorithms such as Iterative Proportional Fitting (IPF) Shi et al. (2023); De Bortoli et al. (2024) can be complex and prone to scalability issues in high dimensions Shi et al. (2023); Bunne et al. (2023); Huang et al. (2024).

To mitigate these complexities, existing research has explored ways to simplify bridge models, for instance, by restricting the model class Kholkin et al. (2024); Song et al. (2020) or employing less precise but faster numerical integration schemes Salimans and Ho (2022); Peyre and Cuturi (2019). While these strategies can improve speed, they often do so at the expense of model expressivity and the quality of the generated samples.

This observation introduces a central tension: the primary motivation for adopting bridge methods is to accelerate sample generation, yet the resulting systems of stochastic differential equations (SDEs) remain difficult to integrate efficiently. An overparameterized deep neural network often represents the drift function in a theoretically optimal bridge. Consequently, most of the model's complexity, interpretability burden, and computational cost reside in this learned network rather than in the bridge formalism itself. A further limitation is that the mathematical form of the bridge SDE is typically fixed by theory, even when a leaned parameterization could suffice. Hence, there is a clear motivation to design alternative bridge formulations that retain accuracy while offering lighter, task-adaptive dynamics.

To address this tension, we propose using surrogate modeling. Our core idea is to learn simplified, data-driven approximations of the underlying dynamics as symbolic differential equations. These surrogates are intended to be (a) simpler and faster to simulate than full-scale neural SDEs and (b) more flexible because they are not strictly bound to a single theoretical form. Conceptually, surrogate models can be trained on trajectories generated by either classical discrete-time diffusion processes or their continuous-time bridge counterparts, providing a versatile pathway to compact symbolic representations. In addition, symbolic discovery methods allow the construction of compact models directly from first principles.

We instantiate this idea with two algorithms. First, SINDy Flow Matching (SINDy-FM) leverages the Sparse Identification of Non-linear Dynamics (SINDy) framework to fit an interpretable, time-dependent dynamical model from either recorded discrete distribution transitions or an online data stream. Second, we reformulate the Schrödinger Bridge problem using Neural Ordinary Differential Equations (Neural ODEs), providing a flexible continuous-time parameterization that we pretrain on physically meaningful trajectories. These symbolic and physics-informed approaches enable the incorporation of domain knowledge, yield highly interpretable models, require far fewer parameters, and can be integrated with high-order, symbol-aware solvers that exhibit faster convergence and superior numerical properties compared to standard SDE solvers.

This methodology points to a new class of bridge models that retain the theoretical advantages of continuous-time dynamics while being parameterized by compact, efficient surrogates. By reducing the parameter count and liberating the model form, we pave the way for creating more diverse, self-adjusting bridges that can better capture the nuances of complex data distributions while remaining computationally tractable for practical deployment. In this paper, we provide background on diffusion and bridge models, present our SINDy-FM and Neural ODE-based methods, and demonstrate their efficacy and efficiency through experiments on Gaussian transport tasks and the MNIST dataset.

## 2 BACKGROUND

### 2.1 DIFFUSION-BASED APPROACH MECHANISM

The field of generative modeling has witnessed significant advances through the development of diffusion-based approaches, including diffusion probabilistic models Sohl-Dickstein et al. (2015), noise-conditioned score networks Yang and Ermon (2019), and denoising diffusion probabilistic models Ho et al. (2020). These methods share theoretical foundations while differing in specific implementations. The forward diffusion process considers a data point sampled from the true data distribution $\mathbf{x}_0 \sim q(\mathbf{x})$ and defines a Markov chain that progressively adds Gaussian noise over $T$ steps, generating a sequence of noisy samples $\mathbf{x}_1, \ldots, \mathbf{x}_T$. The process is governed by a variance schedule $\{\beta_t \in (0, 1)\}\_t = 1^T$, where each transition follows:

$$q(\mathbf{x}_t|\mathbf{x}_{t-1}) = \mathcal{N}(\mathbf{x}_t; \sqrt{1 - \beta_t}\mathbf{x}_{t-1}, \beta_t\mathbf{I}), \quad q(\mathbf{x}_{1:T}|\mathbf{x}_0) = \prod_{t=1}^{T} q(\mathbf{x}_t|\mathbf{x}_{t-1}) \tag{1}$$

As $t$ increases, the sample $\mathbf{x}_0$ undergoes progressive degradation of its distinctive characteristics. In the asymptotic limit $T \to \infty$, $\mathbf{x}_T$ converges to an isotropic Gaussian distribution. The forward process admits an efficient sampling mechanism via reparameterization. Defining $\alpha_t = 1 - \beta_t$ and $\bar{\alpha}_t = \prod_{i=1}^{t} \alpha_i$, the state at any arbitrary time step $t$ can be expressed as:

$$
\begin{aligned}
\mathbf{x}_t = \sqrt{\alpha_t}\mathbf{x}_{t-1} + \sqrt{1 - \alpha_t}\epsilon_{t-1}, \quad & \epsilon_{t-1}, \epsilon_{t-2}, \cdots \sim \mathcal{N}(0, I) \\
& = \sqrt{\alpha_t\alpha_{t-1}}\mathbf{x}_{t-2} + \sqrt{1 - \alpha_t\alpha_{t-1}}\bar{\epsilon}_{t-2} = \sqrt{\bar{\alpha}_t}\mathbf{x}_0 + \sqrt{1 - \bar{\alpha}_t}\epsilon \quad (2)
\end{aligned}
$$

This yields the conditional distribution:

$$
q(\mathbf{x}_t|\mathbf{x}_0) = \mathcal{N}(\mathbf{x}_t; \sqrt{\bar{\alpha}_t}\mathbf{x}_0, (1 - \bar{\alpha}_t)I) \tag{3}
$$

The merging of Gaussian distributions follows standard composition rules: for $\mathcal{N}(0, \sigma_1^2 I)$ and $\mathcal{N}(0, \sigma_2^2 I)$, the resultant distribution is $\mathcal{N}(0, (\sigma_1^2 + \sigma_2^2)I)$. The composite standard deviation is given by:

$$
\sqrt{(1 - \alpha_t) + \alpha_t(1 - \alpha_{t-1})} = \sqrt{1 - \alpha_t\alpha_{t-1}} \tag{4}
$$

Typically, the variance schedule follows a monotonically increasing pattern $\beta_1 < \beta_2 < \cdots < \beta_T$, consequently yielding $\bar{\alpha}_1 > \cdots > \bar{\alpha}_T$. The reverse diffusion process reconstructs samples by reversing the diffusion trajectory, sampling from $q(\mathbf{x}_{t-1}|\mathbf{x}_t)$ to recover data points from Gaussian noise inputs $\mathbf{x}_T \sim \mathcal{N}(0, \mathbf{I})$. Under the condition of sufficiently small $\beta_t$, $q(\mathbf{x}_{t-1}|\mathbf{x}_t)$ remains Gaussian. However, direct estimation of this reverse distribution is computationally intractable as it necessitates integration over the entire dataset. Consequently, a parametric model $p_\theta$ must be learned to approximate these conditional probabilities. The reverse diffusion process is modeled as $p_\theta(\mathbf{x}_{t-1}|\mathbf{x}_t) = \mathcal{N}(\mathbf{x}_{t-1}; \boldsymbol{\mu}_\theta(\mathbf{x}_t, t), \boldsymbol{\Sigma}_\theta(\mathbf{x}_t, t))$. The learning objective involves training $\boldsymbol{\mu}_\theta$ to predict the target:

$$
\hat{\boldsymbol{\mu}}_t = \frac{1}{\sqrt{\alpha_t}}\left(\mathbf{x}_t - \frac{1 - \alpha_t}{\sqrt{1 - \bar{\alpha}_t}}\boldsymbol{\epsilon}_t\right) \tag{5}
$$

Leveraging the availability of $\mathbf{x}_t$ during training, the reparameterization approach facilitates prediction of $\boldsymbol{\epsilon}_t$:

$$
\boldsymbol{\mu}_\theta(\mathbf{x}_t, t) = \frac{1}{\sqrt{\alpha_t}}\left(\mathbf{x}_t - \frac{1 - \alpha_t}{\sqrt{1 - \bar{\alpha}_t}}\boldsymbol{\epsilon}_\theta(\mathbf{x}_t, t)\right) \tag{6}
$$

$$
\mathbf{x}_{t-1} = \mathcal{N}\left(\mathbf{x}_{t-1}; \frac{1}{\sqrt{\alpha_t}}\left(\mathbf{x}_t - \frac{1 - \alpha_t}{\sqrt{1 - \bar{\alpha}_t}}\boldsymbol{\epsilon}_\theta(\mathbf{x}_t, t)\right), \boldsymbol{\Sigma}_\theta(\mathbf{x}_t, t)\right) \tag{7}
$$

The training loss $L_t$ minimizes the discrepancy from $\hat{\boldsymbol{\mu}}$:

$$
\begin{aligned}
L_t &= \mathbb{E}_{\mathbf{x}_0, \boldsymbol{\epsilon}}\left[\frac{1}{2\|\boldsymbol{\Sigma}_\theta(\mathbf{x}_t, t)\|_2^2}\|\hat{\boldsymbol{\mu}}_t(\mathbf{x}_t, \mathbf{x}_0) - \boldsymbol{\mu}_\theta(\mathbf{x}_t, t)\|^2\right] \\
&= \mathbb{E}_{\mathbf{x}_0, \boldsymbol{\epsilon}}\left[\frac{1}{2\|\boldsymbol{\Sigma}_\theta\|_2^2}\left\|\frac{1}{\sqrt{\alpha_t}}\left(\mathbf{x}_t - \frac{1 - \alpha_t}{\sqrt{1 - \bar{\alpha}_t}}\boldsymbol{\epsilon}_t\right) - \frac{1}{\sqrt{\alpha_t}}\left(\mathbf{x}_t - \frac{1 - \alpha_t}{\sqrt{1 - \bar{\alpha}_t}}\boldsymbol{\epsilon}_\theta(\mathbf{x}_t, t)\right)\right\|^2\right] \\
&= \mathbb{E}_{\mathbf{x}_0, \boldsymbol{\epsilon}}\left[\frac{(1 - \alpha_t)^2}{2\alpha_t(1 - \bar{\alpha}_t)\|\boldsymbol{\Sigma}_\theta\|_2^2}\|\boldsymbol{\epsilon}_t - \boldsymbol{\epsilon}_\theta(\mathbf{x}_t, t)\|^2\right] \\
&= \mathbb{E}_{\mathbf{x}_0, \boldsymbol{\epsilon}}\left[\frac{(1 - \alpha_t)^2}{2\alpha_t(1 - \bar{\alpha}_t)\|\boldsymbol{\Sigma}_\theta\|_2^2}\|\boldsymbol{\epsilon}_t - \boldsymbol{\epsilon}_\theta(\sqrt{\bar{\alpha}_t}\mathbf{x}_0 + \sqrt{1 - \bar{\alpha}_t}\boldsymbol{\epsilon}_t, t)\|^2\right] \quad (8)
\end{aligned}
$$

## 2.2 Diffusion bridges

The Schrödinger Bridge (SB) Shi et al. (2023); Peyre and Cuturi (2019)problem addresses the fundamental task of finding a stochastic dynamic mapping between two probability distributions $\pi_0$ and $\pi_T$ by identifying a path measure $\mathbb{P}^{\text{SB}}$ that minimizes the Kullback-Leibler divergence to a reference measure $\mathbb{Q}$ while satisfying marginal constraints:

$$\mathbb{P}^{\text{SB}} = \underset{\mathbb{P}}{\arg\min} \left\{ \text{KL}(\mathbb{P} \mid \mathbb{Q}) : \mathbb{P}_0 = \pi_0, \mathbb{P}_T = \pi_T \right\}.$$

When $\mathbb{Q}$ represents a Brownian motion, this formulation yields an entropy-regularized optimal transport solution. Traditional approaches like Iterative Proportional Fitting (IPF) alternate between projections based on marginal constraints but suffer from error accumulation and scalability issues in high dimensions. To overcome these limitations, we introduce Iterative Markovian Fitting (IMF), a novel methodology that alternately projects onto the space of Markov processes $\mathcal{M}$ and the reciprocal class $\mathcal{R}(\mathbb{Q})$:

$$\mathbb{P}^{2n+1} = \text{proj}_{\mathcal{M}}(\mathbb{P}^{2n}), \quad \mathbb{P}^{2n+2} = \text{proj}_{\mathcal{R}(\mathbb{Q})}(\mathbb{P}^{2n+1}).$$

Unlike IPF, IMF preserves initial and terminal distributions at each iteration. Building on IMF, we propose Diffusion Schrödinger Bridge Matching (DSBM), a scalable algorithm that combines forward and backward Markovian projections with efficient bridge sampling. DSBM leverages simple regression losses akin to Bridge Matching and achieves superior performance in approximating SB solutions across various transport tasks, including high-dimensional domains and image translation, while mitigating bias accumulation via symmetric forward-backward updates.

## 3 Proposed approaches

### 3.1 SINDy Flow Matching (SINDy-FM)

We propose *SINDy Flow Matching (SINDy-FM)*, a generative algorithm that constructs supervised samples of a target vector field from an interpolation between endpoint distributions and fits an interpretable time-dependent dynamical model through sparse identification of non-linear dynamics (SINDy) Brunton et al. (2016) by loss of flow matching style Lipman et al. (2023). The method learns a velocity field $v_\theta(x, t)$ that, when integrated from $t = 0$ to $1$, transports an initial sample $x(0) \sim p_0$ to a terminal state whose distribution approximates $p_1$.

Let $p_0$ and $p_1$ be probability measures on $\mathbb{R}^d$. We seek a time-varying field $v_\theta : \mathbb{R}^d \times [0, 1] \to \mathbb{R}^d$ such that the ODE

$$\frac{d}{dt}x(t) = v_\theta\big(x(t), t\big), \qquad x(0) \sim p_0, \tag{9}$$

induces a terminal distribution $x(1)$ that matches $p_1$.

Given an interpolation $\gamma : [0, 1] \times \mathbb{R}^d \times \mathbb{R}^d \to \mathbb{R}^d$ that connects endpoints $x_0 \sim p_0$ and $x_1 \sim p_1$, we define supervision pairs by

$$x(t) = \gamma(t; x_0, x_1), \qquad \dot{x}(t) = \partial_t \gamma(t; x_0, x_1).$$

Sampling independent $(x_0, x_1)$ pairs and times $t \in [0, 1]$ (optionally $m > 1$ samples of $t$ for each combination of $(x_0, x_1)$) yields a dataset $\mathcal{D} = \{(x_i, t_i, \dot{x}_i)\}_{i=1}^{N}$ of states, time stamps, and *exact* time derivatives. This bypasses numerical differentiation, which is typical for the identification of differential equation systems from raw data, and provides low-noise targets for regression of the field.

We represent $v_\theta$ as a sparse linear combination of $p$ features in the form of symbolic expressions. Let $\Xi(x, t) \in \mathbb{R}^p$, and write $v_\theta(x, t) = W \Xi(x, t)$ with coefficient matrix $W \in \mathbb{R}^{d \times p}$. We estimate $W$ using sparse regression.

Given $\mathcal{D}$, parameters are obtained by minimizing the mean-squared discrepancy between predicted and supervised derivatives:

$$\min_W \ \frac{1}{N} \sum_{i=1}^{N} \big\| W \Xi(x_i, t_i) - \dot{x}_i \big\|_2^2 \ + \ \text{(sparsity loss)}. \tag{10}$$

The minimizer of this loss, according to the flow-matching approach, recovers the marginals $p_t$ of the interpolant. This is also the loss traditionally used in SINDy for approximating trajectories that in this setting would be formed by $\gamma(; x_0, x_1)$ for the fixed endpoints. Whereas sampling more than one $t$ per trajectory slightly reduces space coverage due to correlated samples, it empirically stabilizes training in some scenarios by allowing the model to identify dependencies within the trajectory.

After fitting, transport is effected by integrating equation 9 for fresh $x_0 \sim p_0$ from $t = 0$ to $t = 1$. The collection of terminal states $\{x(1)\}$ constitutes samples from the model-implied terminal distribution.

The algorithm is presented in Figure 3.1.

[t] SINDy Flow Matching (SINDY-FM) Distributions $p_0, p_1$; interpolation $\gamma(t; x_0, x_1)$; state library $\Phi$, optional time library $\Psi$; integers $N$ (trajectories) and $m$ (time points per trajectory). $\mathcal{D} \leftarrow \varnothing$ $i = 1$ $N$ Sample $x_0 \sim p_0$, $x_1 \sim p_1$ Draw and sort $t_{i1}, \ldots, t_{im} \sim \mathrm{U}[0,1]$ $j = 1$ $m$ $x_{ij} \leftarrow \gamma(t_{ij}; x_0, x_1)$ $\dot{x}_{ij} \leftarrow \partial_t \gamma(t_{ij}; x_0, x_1)$ Add $(x_{ij}, t_{ij}, \dot{x}_{ij})$ to $\mathcal{D}$ Compute features $\Xi(x, t)$ on the points $(x_i, t_i)$ of the dataset Fit SINDy on $\mathcal{D}$ with loss (10) to approximate $\dot{x}_i$ obtaining $v_\theta(x, t) = W \Xi(x, t)$ **Deployment:** For any $x_0 \sim p_0$, solve $\dot{x} = v_\theta(x, t)$, $x(0) = x_0$, to $t = 1$ and return $x(1)$

## 3.2 DIFFUSION SCHRÖDINGER BRIDGE MATCHING–NEURALODE (DSBM-NEURALODE)

We first implement a Denoising Diffusion Probabilistic Model (DDPM) Nichol and Dhariwal (2021), training a U-Net to predict the noise added during the forward diffusion process. This process can be described as a forward diffusion process governed by the following SDE

$$dX_t = -\frac{1}{2}\beta(t)X_t dt + \sqrt{\beta(t)}dB_t \tag{11}$$

where $X_t \in \mathbb{R}^d$ is the system state at time $t \in [0, 1]$, $\beta(t)$ is a predefined schedule function with $\beta(t) = \beta_{\text{start}} + (\beta_{\text{end}} - \beta_{\text{start}}) \cdot t$, and $B_t$ is standard Brownian motion in $\mathbb{R}^d$. The continuous process is discretized into $N$ steps. For discrete time $t \in \{0, 1, \ldots, N\}$, the state evolves as:

$$X_t = \sqrt{\bar{\alpha}_t}X_0 + \sqrt{1 - \bar{\alpha}_t}\epsilon \tag{12}$$

where $\bar{\alpha}_t = \prod_{s=1}^{t}(1 - \beta_s)$ is the cumulative product and $\epsilon \sim \mathcal{N}(0, \mathbf{I}_d)$ is standard Gaussian noise. The U-Net parameterized by $\theta$ learns to predict the noise $\epsilon$:

$$\mathcal{L}_{\text{DDPM}}(\theta) = \mathbb{E}_{t \sim \mathcal{U}[0,N], X_0 \sim p_0, \epsilon \sim \mathcal{N}(0,\mathbf{I})}\left[\|\epsilon - \epsilon_\theta(X_t, t)\|_2^2\right] \tag{13}$$

where $\epsilon_\theta(X_t, t)$ is the U-Net prediction.

After training, the generation uses the reverse SDE:

$$X_{t-1} = \frac{1}{\sqrt{\alpha_t}}\left(X_t - \frac{1 - \alpha_t}{\sqrt{1 - \bar{\alpha}_t}}\epsilon_\theta(X_t, t)\right) + \sigma_t \mathbf{z} \tag{14}$$

where $\mathbf{z} \sim \mathcal{N}(0, \mathbf{I}_d)$ and $\sigma_t^2 = \beta_t$.

Let $\{X_{t_k}^{(i)}\}\_k = 0^m$ for $i = 1, \ldots, N$ denote trajectories sampled from this reference diffusion process. Each trajectory has shape (batch size, num steps $+ 1$, dim), where the batch dimension indexes independent trajectories, the temporal axis enumerates discrete time points from $t = 0$ to $t = 1$, and dim is the state dimension. We construct the training dataset from pairs of consecutive states: $\mathcal{D} = \{(X_{t_k}^{(i)}, X_{t_{k+1}}^{(i)}) : i = 1, \ldots, N, \ k = 0, \ldots, m - 1\}$.

The Schrödinger Bridge problem is defined as the solution to the minimization problem: $\mathbb{P}^{SB} = \arg\min \mathrm{KL}(\mathbb{P}|\mathbb{Q}) : \mathbb{P}_0 = \pi_0, \mathbb{P}_T = \pi_T$, where $\mathbb{Q}$ is a reference measure on path space $\mathcal{C} = C([0, T], \mathbb{R}^d)$, and KL denotes the Kullback-Leibler divergence. In the dynamic formulation, the reference measure $\mathbb{Q}$ is typically given by a stochastic differential equation: $dX_t = f_t(X_t)dt + \sigma_t dB_t$, with $X_0 \sim \pi_0$, where $(B_t)$ is a Wiener process and $\sigma_t > 0$ controls the diffusion strength.

The DSBM Shi et al. (2023) approach introduces an iterative procedure that alternates between projections onto two fundamental sets: the space of Markov processes $\mathcal{M}$ and the reciprocal class

$\mathcal{R}(\mathbb{Q})$ consisting of measures sharing the same bridge as the reference measure. The iterative sequence is defined by:

$$\mathbb{P}^{2n+1} = \text{proj}_{\mathcal{M}}(\mathbb{P}^{2n}), \quad \mathbb{P}^{2n+2} = \text{proj}_{\mathcal{R}(\mathbb{Q})}(\mathbb{P}^{2n+1})$$

with initialization satisfying $\mathbb{P}_0^0 = \pi_0$ and $\mathbb{P}_T^0 = \pi_T$. For a mixture of bridges $\Pi = \Pi_{0,T}\mathbb{Q}[0,T]$, the Markovian projection $\mathbb{M}^* = \text{proj}_{\mathcal{M}}(\Pi)$ follows the SDE

$$d\mathbf{X}_t^* = \left[f_t(\mathbf{X}_t^*) + v_t^*(\mathbf{X}_t^*)\right]dt + \sigma_t\,d\mathbf{B}_t,$$

where the optimal drift is given by

$$v_t^*(x_t) = \sigma_t^2\,\mathbb{E}_{\Pi_{T|t}}\left[\nabla \log \mathbb{Q}_{T|t}(\mathbf{X}_T \mid \mathbf{X}_t) \,\big|\, \mathbf{X}_t = x_t\right]. \tag{15}$$

DSBM with Neural ODE presents a methodology for solving the Schrödinger Bridge problem by integrating continuous-time neural ordinary differential equations (Neural ODEs) with iterative Markovian fitting. Our approach leverages pre-training on diffusion trajectories to initialize the bridge dynamics. This DSBM-NeuralODE method parameterizes the drift in continuous time with Neural ODEs Chen et al. (2018). For each direction $d \in$ forward, backward, we define a neural network $f_\theta^d : \mathbb{R} \times \mathbb{R}^n \to \mathbb{R}^n$ that takes time $t$ and state $\mathbf{x}$ as input and outputs the drift vector. The drift function for direction $d$ is given by $v_\theta^d(t, \mathbf{x}) = f_\theta^d(t, \mathbf{x})$, where $f_\theta^d$ is implemented as a feedforward neural network $\text{ODEFunc}_\theta : \mathbb{R}^d \times [0, 1] \to \mathbb{R}^d$

$$\frac{d\mathbf{X}_t}{dt} = \text{ODEFunc}_\theta(\mathbf{X}_t, t) \tag{16}$$

For the forward network, we minimize:

$$\mathcal{L}_{\text{forward}}(\theta) = \mathbb{E}_{(\mathbf{X}_t, \mathbf{X}_{t+\Delta t}) \sim \mathcal{D}}\left[\left\|\mathbf{X}_{t+\Delta t} - \left(\mathbf{X}_t + f_\theta^{\text{forward}}(t, \mathbf{X}_t) \cdot \Delta t\right)\right\|^2\right] \tag{17}$$

For the backward network:

$$\mathcal{L}_{\text{backward}}(\phi) = \mathbb{E}_{(\mathbf{X}_t, \mathbf{X}_{t+\Delta t}) \sim \mathcal{D}}\left[\left\|\mathbf{X}_t - \left(\mathbf{X}_{t+\Delta t} + f_\phi^{\text{backward}}(t, \mathbf{X}_{t+\Delta t}) \cdot (-\Delta t)\right)\right\|^2\right] \tag{18}$$

This pre-training initializes the networks to approximate the dynamics of the reference process, providing a warm start for the Schrödinger Bridge optimization. Given endpoint pairs $(\mathbf{z}_0, \mathbf{z}_1)$ and $t \sim U(\epsilon, 1-\epsilon)$, we generate training points through Brownian bridge interpolation:

$$\mathbf{z}_t = t\mathbf{z}_1 + (1-t)\mathbf{z}_0 + \sigma\sqrt{t(1-t)}\boldsymbol{\epsilon} \tag{19}$$

For the initial iteration we adopt the reference coupling strategy $\mathbf{X}_1 = \mathbf{X}_0 + \sigma\mathbf{Z}$ with $\mathbf{Z} \sim \mathcal{N}(0, I)$. The training objective for direction $d$ at iteration $n$ is

$$\mathcal{L}_n^d(\theta) = \mathbb{E}_{(\mathbf{X}_0, \mathbf{X}_1) \sim \Pi_{0,T}^n}\mathbb{E}_{t \sim U[\epsilon, 1-\epsilon]}\mathbb{E}_{\mathbf{Z} \sim \mathcal{N}(0, I)}\left[\left\|\mathbf{v}_{\text{target}}^d - f_\theta^d(t, \mathbf{X}_t)\right\|^2\right] \tag{20}$$

where $\mathbf{X}_t$ is computed via Brownian bridge interpolation. We implement the Euler-Maruyama scheme as the sampling algorithm, discretizing the SDE $d\mathbf{X}_t = f_\theta^d(t, \mathbf{X}_t)dt + \sigma d\mathbf{B}_t$.

## 4 EXPERIMENTAL EVALUATION

### 4.1 EXPERIMENTAL SETTINGS

We study two problems: (i) a family of Gaussian-to-Gaussian transport in moderate dimension and (ii) latent translation between class-conditional MNIST manifolds. We compare SINDy-FM and DSBM-NeuralODE with a traditional NN-based DSBM baseline. Unless specified otherwise, all endpoints

live in five-dimensional Euclidean space for the Gaussian benchmarks and in the eight-dimensional latent space induced by a convolutional variational autoencoder for MNIST.

The DSBM baseline in both the Gaussian and MNIST settings is trained on the same CPU using the identical latent pairs, with the drift parameterized by a feed-forward MLP (hidden width 64 for Gaussians, 128 for MNIST, depth 2) and a diffusion strength of $\sigma = 1$.

### 4.1.1 GAUSSIAN TRANSPORT.

We generate source and target marginals $p_0 = \mathcal{N}(\mu_0, \Sigma_0)$ and $p_1 = \mathcal{N}(-\mu_0, \Sigma_1)$ across five representative covariance scenarios: identity, diagonal, rotated, high_condition, and asymmetric (with differing eigenvalue scales for $\Sigma_0$ and $\Sigma_1$). The identity scenario is evaluated for multiple problem dimensions and distribution means. Covariance spectra are sampled log-uniformly within prescribed ranges and, in the rotated and asymmetric cases, conjugated by random orthogonal matrices. For each pair, we draw $5 \times 10^4$ straight-line flow-matching trajectories, each with two uniformly sampled time stamps – the supervision couples state samples with exact time derivatives along the linear path. We evaluate candidate fields by integrating from $t = 0$ to 1 and reporting the $W_2$ Wasserstein distance, training time, inference time, and number of learnable parameters for the studied methods: SINDy-FM, DSBM-NeuralODE, and the neural baseline–DSBM Shi et al. (2023).

For SINDy-FM in gaussian case, we parameterize the drift as $v(x, t) = K(t), x + k(t)$ with $K(t) \in \mathbb{R}^{d \times d}$ and $k(t) \in \mathbb{R}^d$, motivated by the conditional Gaussian construction in Flow Matching where the canonical vector field is affine in $x$ Theorem 3 in Lipman et al. (2023). An affine drift preserves Gaussianity along the flow (since $\dot{\mu}_t = K(t)\mu_t + k(t)$ and $\dot{\Sigma}_t = K(t)\Sigma_t + \Sigma_t K(t)^\top$), and for Gaussian endpoints the population FM minimizer is itself affine (under OT/Monge or independent pairings).

In the Gaussian case, SINDy-FM parameterizes the drift as $v(x, t) = K(t)x + k(t)$ with $K(t) \in \mathbb{R}^{d \times d}$ and $k(t) \in \mathbb{R}^d$, motivated by the conditional Gaussian construction in flow matching where the canonical vector field is affine in $x$ (Theorem 3 in Lipman et al. (2023)). An affine drift preserves Gaussianity along the flow (because $\dot{\mu}_t = K(t)\mu_t + k(t)$ and $\dot{\Sigma}_t = K(t)\Sigma_t + \Sigma_t K(t)^\top$), and for Gaussian endpoints the population FM minimizer is itself affine under either OT/Monge or independent pairings.

In practice, we expand the time dependence in a polynomial basis,

$$K(t) = \sum_{r=0}^{R} A_r t^r, \qquad k(t) = \sum_{r=0}^{R} b_r t^r, \tag{21}$$

with learnable coefficients $A_r \in \mathbb{R}^{d \times d}$ and $b_r \in \mathbb{R}^d$. Such polynomial bases are a standard SINDy choice and are universal approximators on compact intervals, so the features follow the factorization $\Xi(x, t) = [\Phi(x) \otimes \Psi(t)]$. In the identity scenario for dimensions 20 and 50, we reduce the polynomial order in $t$ to 1, preserving the true constant drift for that class. The polynomial order $R$ chosen for experimentation is 2. We optimize with SR3 using threshold 0.10, $\nu = 10^{-2}$, and two time samples per trajectory, and we perform $K = 20$ integration steps at inference.

For each method, a convergence test was performed to determine the optimal number of function evaluations. The example of convergence analysis is shown in Figure 1.

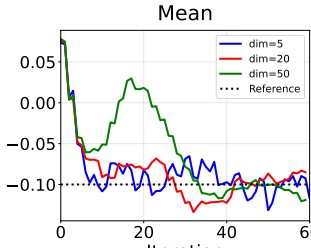 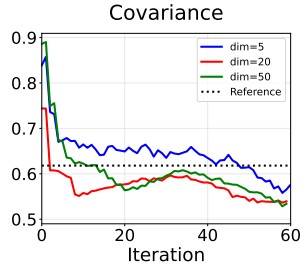 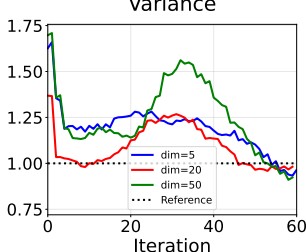

Figure 1: DSBM-NeuralODE convergence in Gaussian experiment for mean=0.1

Table 1 shows varying mean scale from 0.1 to 10 and lifting the dimension from 5 to 50. The SINDy-FM library keeps $W_2$ essentially flat. Crucially, per-sample inference remains in the microsecond range throughout. At the same time, the neural DSBM baseline requires two orders of magnitude more time and sees its transport error climb rapidly under the same conditioning.

Table 1: Comparison of Three Methods Across Different Distribution Parameters

| Mean | Dim | Method | W2 | Train Time (s) | Inference Time (s) | Parameters |
|---|---|---|---|---|---|---|
| | | Gaussian SINDy-FM | 0.172 | 27.76 | $8 \cdot 10^{-6}$ | 25 |
| 0.1 | 5 | DSBM-NeuralODE | 0.131 | 2326.65 | 21.79 | $2.7 \cdot 10^5$ |
| | | DSBM | 0.103 | 90 | 0.08 | $4.9 \cdot 10^3$ |
| | | Gaussian SINDy-FM | 0.128 | 46.74 | $5 \cdot 10^{-6}$ | 20 |
| 0.1 | 20 | DSBM-NeuralODE | 0.170 | 2440.33 | 31.00 | $1.1 \cdot 10^6$ |
| | | DSBM | 0.220 | 260 | 0.08 | $6.9 \cdot 10^3$ |
| | | Gaussian SINDy-FM | 0.303 | 49.25 | $1.5 \cdot 10^{-5}$ | 50 |
| 0.1 | 50 | DSBM-NeuralODE | 0.219 | 2450.53 | 29.64 | $1.2 \cdot 10^6$ |
| | | DSBM | 0.450 | 570 | 0.19 | $3.0 \cdot 10^4$ |
| | | Gaussian SINDy-FM | 0.167 | 27.66 | $8 \cdot 10^{-6}$ | 29 |
| 1 | 5 | DSBM-NeuralODE | 0.144 | 3098.43 | 32.76 | $2.7 \cdot 10^5$ |
| | | DSBM | 0.066 | 110 | 0.06 | $4.9 \cdot 10^3$ |
| | | Gaussian SINDy-FM | 0.128 | 49.61 | $5 \cdot 10^{-6}$ | 20 |
| 1 | 20 | DSBM-NeuralODE | 0.179 | 3157.54 | 45.52 | $1.1 \cdot 10^6$ |
| | | DSBM | 0.220 | 300 | 0.09 | $6.9 \cdot 10^3$ |
| | | Gaussian SINDy-FM | 0.303 | 50.38 | $1.5 \cdot 10^{-5}$ | 50 |
| 1 | 50 | DSBM-NeuralODE | 0.212 | 3374.63 | 48.39 | $1.2 \cdot 10^6$ |
| | | DSBM | 0.470 | 550 | 0.21 | $3.0 \cdot 10^4$ |
| | | Gaussian SINDy-FM | 0.167 | 34.97 | $9 \cdot 10^{-6}$ | 27 |
| 10 | 5 | DSBM-NeuralODE | 0.175 | 5754.54 | 71.13 | $2.7 \cdot 10^5$ |
| | | DSBM | 0.110 | 280 | 0.06 | $4.9 \cdot 10^3$ |
| | | Gaussian SINDy-FM | 0.128 | 48.45 | $5 \cdot 10^{-6}$ | 20 |
| 10 | 20 | DSBM-NeuralODE | 0.215 | 5934.34 | 80.87 | $1.1 \cdot 10^6$ |
| | | DSBM | 0.260 | 520 | 0.14 | $2.2 \cdot 10^4$ |
| | | Gaussian SINDy-FM | 0.303 | 50.36 | $1.5 \cdot 10^{-5}$ | 50 |
| 10 | 50 | DSBM-NeuralODE | 0.247 | 6027.98 | 82.12 | $1.2 \cdot 10^6$ |
| | | DSBM | 0.550 | 750 | 0.41 | $9.2 \cdot 10^4$ |

Across the remaining covariance scenarios in Table 2, SINDy-FM again matches or beats the neural Schrödinger-bridge baselines yet remains vastly faster. Identity and diagonal experiments differ by only a few hundredths in $W_2$ versus the best neural score. However, the symbolic drift integrates two to three orders of magnitude faster (microseconds instead of milliseconds). The asymmetric and high_condition benchmarks underscore the efficiency gap: SINDy-FM preserves $W_2 \approx 0.25$ without instability, whereas DSBM must run considerably longer to approach similar accuracy.

Table 2: Comparison of Gaussian SINDy-FM and DBSM Methods Across Different Scenarios

| Scenario | Method | W2 | Train Time/s | Inference Time/s | Parameters |
|---|---|---|---|---|---|
| identity | Gaussian SINDy-FM | 0.167 | 29.18 | $8 \cdot 10^{-6}$ | 29 |
| | DBSM | 0.12 | 150 | 0.1 | $4.9 \cdot 10^3$ |
| diagonal | Gaussian SINDy-FM | 0.162 | 30.61 | $8 \cdot 10^{-6}$ | 71 |
| | DBSM | 0.2/0.12 | 190 | 0.04 | $4.9 \cdot 10^3$ |
| rotated | Gaussian SINDy-FM | 0.220 | 31.27 | $8 \cdot 10^{-6}$ | 76 |
| | DBSM | 0.21/0.15 | 180 | 0.06 | $4.9 \cdot 10^3$ |
| asymmetric | Gaussian SINDy-FM | 0.254 | 31.40 | $8 \cdot 10^{-6}$ | 81 |
| | DBSM | 0.7/2.3 | 1100 | 0.05 | $4.9 \cdot 10^3$ |
| high_condition | Gaussian SINDy-FM | 0.233 | 30.84 | $8 \cdot 10^{-6}$ | 85 |
| | DBSM | 1.9/2.2 | 900 | 0.06 | $4.9 \cdot 10^3$ |

### 4.1.2 MNIST LATENT TRANSLATION

For MNIST, we translate digit-2 latent codes into digit-3 latents using a convolutional VAE encoder (trained to 99% reconstruction accuracy) and a lightweight classifier (99.9% validation accuracy) to monitor digit purity. The flow-matching supervision consists of $1.2 \times 10^5$ latent pairs with three uniformly sampled time points per pair; training batches interleave SINDy-FM and the neural Schrödinger-bridge baseline (DSBM). The SINDy-FM translator employs the "rich" library obtained by the tensor product $\Phi(x) \otimes \Psi(t)$, where $\Psi(t) = \{1, t, t^2\}$ and $\Phi(x)$ includes all monomials in the latent coordinates up to degree two, so the learned drift is quadratic in $x$ with polynomial time dependence.

On MNIST (Figure 3), SINDy-FM enables control over the quality–complexity trade-off through the choice of sparsity penalization $\nu$ and threshold, while maintaining comparable generation quality. The denser variant demonstrates 92% digit-3 accuracy (as classified by a near-ideal CNN). It achieves a Fréchet Inception Distance of 83 and an Inception Score of 1.4 in the feature space of the MNIST-trained classifier. DSBM attains similar metrics but has slower inference and requires three orders of magnitude more parameters.

Table 3: MNIST latent translation comparison

| Metric | SINDy-FM | | DSBM |
|---|---|---|---|
| | thresh=0.02, $\nu = 0.05$ | thresh=0.05, $\nu = 0.1$ | |
| Digit accuracy | $0.919 \pm 0.010$ | 0.914 | 0.912 |
| FID | $83.48 \pm 9.98$ | $89.38 \pm 7.78$ | 72.17 |
| IS | $1.431 \pm 0.060$ | $1.466 \pm 0.056$ | 1.47 |
| Train time/s | 41 | 24.68 | 450 |
| Inference time/s | $9.34 \cdot 10^{-4}$ | $9.36 \cdot 10^{-4}$ | $8.0 \cdot 10^{-2}$ |
| Active params | 923 | 541 | $2.72 \cdot 10^5$ |

An example of translated 3s generated by SINDy-FM is shown in Figure 2.

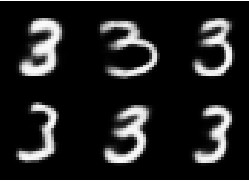

Figure 2: MNIST digit 3 generation via translation from 2s with SINDy-FM

## 5    Discussion

Our empirical findings indicate that surrogate modeling offers a feasible approach to harmonize the theoretical appeal of continuous-time diffusion bridges with the pragmatic requirements of efficient, interpretable sample generation. The introduced SINDy-FM method effectively obtained compact symbolic representations of the transport dynamics. In Gaussian transport scenarios, these surrogates achieved competitive Wasserstein distances while requiring significantly fewer parameters and enabling nearly instantaneous inference. This parsimony offers a notable advantage, yielding models that are not only rapid but also highly interpretable, as the sparse coefficients explicitly reveal the preeminent dynamical structure. For the more intricate MNIST latent translation task, the symbolic translator preserved robust performance in digit-classification accuracy and low latent-space FID, substantiating that the methodology can scale to semantically rich, non-linear manifolds.

The principal strength of symbolic surrogates such as SINDy-FM resides in their data efficiency and transparency. When the underlying dynamics are well-represented by the selected feature library, these models can be precisely identified from relatively few samples. The resultant white-box models facilitate diagnostics, incorporate physical constraints, and exhibit robust extrapolation behavior. Nonetheless, this approach is intrinsically constrained by the expressivity of the pre-defined library. A suboptimal basis will limit the model's accuracy, and sparsity-inducing regularization introduces a bias-variance trade-off that requires careful management.

Conversely, the DSBM-NeuralODE method represents a distinct class of surrogate that balances flexibility with the continuous-time framework of the Schrödinger Bridge. By parameterizing the drift function with a Neural ODE, this approach circumvents strict adherence to a fixed theoretical form, enabling the learning of more complex, non-linear dynamics that may be intractable for a simple symbolic library. Pre-training on diffusion trajectories provides a physically meaningful initialization, stabilizing subsequent SB optimization. While this method incurs higher computational costs during training and inference than SINDy-FM and yields a less interpretable black-box model, it serves as a potent and adaptable surrogate. It effectively bridges the gap between the rigid, theory-prescribed bridges and the necessity for adaptable models, particularly in scenarios where the optimal transport dynamics are intricate and not readily captured by a sparse linear combination of basic functions.

## 6    Conclusion

The investigation of surrogate models suggests a more diverse ecosystem of bridge models. Symbolic surrogates excel in scenarios with inherent structure and where interpretability and speed are of utmost importance. Neural surrogates, such as the Neural ODE variant, present a compelling alternative for capturing highly complex dynamics at a higher computational expense. The decision between them hinges on the specific requirements of the task—accuracy, speed, interpretability, and the known structure of the data. Future research will focus on expanding the symbolic toolkit with more complex differential structures and programmatic discovery, while refining neural surrogates for greater efficiency, ultimately paving the way for a new generation of self-adjusting, computationally tractable generative models.

### Acknowledgments

The research was carried out within the state assignment of Ministry of Science and Higher Education of the Russian Federation (project FSER-2024-0004).

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
