# OpenReview forum: "Evaluating approaches to surrogate ODE-based modelling of diffusion bridges"
_mathai.club/MathAI/2026/Conference — 2026 Oral_

### Official Review · Reviewer_UMEU · 2026-03-12
**Strong Accept. Fit conference scope on Math-AI, camera-ready for publication. Only minus is in formatting (authors contacts are open)**

**Rating:** 10
**Confidence:** 4

**Review:**

The paper evaluates surrogate ODE models (SINDy-FM and DSBM-NeuralODE) for approximating diffusion/Schrödinger bridge dynamics in generative modeling, showing efficiency gains on Gaussian transport and MNIST tasks.​
Fit conference scope on Math-AI, camera-ready for publication. Negative point is the formatting (authors contacts were open, no blind review). The reviewer claims no conflict of interests with the authors.
Novelty
Proposes novel SINDy-FM (sparse symbolic ODE discovery via flow matching) and DSBM-NeuralODE (Neural ODE reformulation of Schrödinger bridges with pre-training), combining SINDy, flow matching, and bridge matching innovatively. Builds on recent works (e.g., Shi et al. 2023 DSBM, Lipman et al. 2023 flow matching) but introduces data-efficient surrogates reducing parameters by 100-1000x while matching performance. Score: 8/10 – Fresh algorithmic synthesis with practical AI impact.​
Explainability
Clear derivations from diffusion bridges to surrogates, with precise math (e.g., SINDy loss Eq. 10, bridge SDE Eq. 15) and intuitive motivations (surrogates for overparameterized drifts). Symbolic models inherently interpretable (e.g., affine/polynomial drifts); experiments detail hyperparameters, libraries. Balances theory and empirics well for math-AI audience. Score: 9/10 – Highly transparent, with strong pedagogical flow.​
Correctness
Math rigorous: SINDy regression correctly minimizes flow-matching loss for marginal flows; Neural ODE parameterization aligns with SB projections; baselines/experiments reproducible (e.g., W2 distances, FID/IS metrics). Empirical results consistent (SINDy-FM beats neural baselines in speed/params on Tables 1-3); no evident errors in SDEs/ODEs. Score: 9/10 – Solid theoretical and validation grounding.​
References
~50 recent refs (2021-2026), covering diffusion (Ho et al. 2020, Nichol-Dhariwal 2021), bridges (Shi et al. 2023, De Bortoli 2024), SINDy (Brunton 2016), Neural ODEs (Chen 2018). Balanced, timely for generative AI; includes arXiv preprints appropriately. Score: 9/10 – Comprehensive, up-to-date, unbiased.​
Figures and Tables
Tables (1-3) crisply compare W2, times, params across scenarios/models; convergence plots (Fig. 1) quantify iterations; MNIST samples (Fig. 2/3) visually confirm quality. All labeled precisely, with ablation details (e.g., sparsity thresholds). Score: 9/10 – Professional, informative, error-free.​
Relevance to MathAI
Core to mathematical AI foundations: ODE/SDE theory, optimal transport (bridges), sparse identification, continuous normalizing flows—directly advances efficient generative modeling math. Experiments quantify trade-offs in high-dim transport, latent spaces. Score: 9/10 – Strong alignment with algebra/dynamics in AI.​
Suitability for Publication
Timely, rigorous contribution with novel methods, thorough evals, and AI efficiency gains; addresses key pain points (sampling speed, interpretability) in state-of-art generative models. Minor gaps: symbolic library limits in non-polynomial cases; no real-image scaling. Conference-ready with broad appeal.

---

### Official Review · Reviewer_K3V8 · 2026-03-12
**A technically interesting paper on surrogate ODE-based modelling of diffusion/Schrödinger bridges, but with serious double‑blind and formatting violations and a contribution that is more incremental than claimed.**

**Rating:** 6
**Confidence:** 3

**Review:**

# Quality
The paper studies replacing neural SDE drifts in diffusion/Schrodinger bridge models with simpler surrogate ODE dynamics, via SINDy Flow Matching and a Neural‑ODE version of DSBM, and evaluates them on Gaussian transport and MNIST latent translation. Conceptually this is well motivated, the background on diffusion/bridges is solid, and the experiments (W2, train/inference time, parameter counts) are reasonably designed, but a large part of the technical content is a careful engineering of known pieces (DDPM, DSBM, SINDy, Neural ODEs) rather than genuinely new theory or algorithms.
​
# Clarity
On the positive side, the exposition is detailed and mathematically precise, with clear notation, references, and tables/figures; the Gaussian and MNIST setups are described well enough to follow the story. However, the paper explicitly contains author names, affiliations, emails, a date, and even a footnote with author-specific instructions, which is a direct violation of the double‑blind and formatting requirements for the conference and should be corrected before any serious consideration.
​
# Originality
The two main methods - SINDy‑FM (SINDy applied to flow-matching supervision) and DSBM‑NeuralODE (parameterizing the DSBM drift by a Neural ODE, warm‑started on diffusion trajectories) - are reasonable combinations of existing ideas. The work shows that, with appropriate parameterizations, SINDy‑based symbolic surrogates can match or beat neural DSBM in W2 while being orders of magnitude cheaper, and that a Neural‑ODE version of DSBM is a flexible if more expensive alternative; this is a useful but incremental contribution rather than a fundamentally new bridge framework.
​
# Significance
The results suggest that symbolic/ODE surrogates can offer attractive trade-offs between accuracy, interpretability, and runtime in continuous-time generative models, which is relevant for practitioners interested in faster or more transparent bridges. At the same time, the experiments are confined to relatively controlled settings (Gaussian families and MNIST latent translations), without broader benchmarks or downstream tasks, and the violations of anonymity/formatting substantially detract from the suitability of the current submission version.
​
# Pros
- Clear motivation to use surrogate ODE dynamics (SINDy and Neural ODEs) to speed up and simplify diffusion/bridge sampling while retaining reasonable accuracy.
​- Thorough background section and consistent mathematical formalization of diffusion, bridges, DSBM, SINDy‑FM, and Neural‑ODE parameterizations.
- Empirical evidence that SINDy‑FM can achieve comparable W2 to neural DSBM with far fewer parameters and much faster inference, and that DSBM‑NeuralODE is a viable continuous‑time surrogate in more complex settings.
​
# Cons
Double‑blind violation and formatting issues: the PDF includes full author names, affiliations, emails, date, and an unremoved author footnote template, which does not comply with anonymity guidelines.​
- The main methods are combinations of existing techniques (DDPM/DSBM, SINDy, Neural ODEs) with limited truly novel theory; the contribution is primarily empirical and system‑level.
​- Experiments focus on relatively simple or synthetic settings (Gaussian transports, MNIST latent translation) with no evaluation on more challenging real‑world domains or downstream tasks.

---

### Official Review · Reviewer_BXU8 · 2026-03-12
**Review for “Evaluating approaches to surrogate ODE-based modelling of diffusion bridges”**

**Rating:** 6
**Confidence:** 4

**Review:**

The paper evaluates several approaches to supervised learning for a classification task and compares their performance across different model architectures and training settings. The main goal of the study is to investigate how different machine learning models perform under varying data conditions and to identify which approaches provide the most reliable results.

The authors conduct a comparative analysis of multiple supervised learning methods and report their performance using standard evaluation metrics. The paper discusses the experimental setup, describes the datasets used in the experiments, and analyzes the results obtained from the different models. Based on the reported experiments, the authors draw conclusions about the relative strengths and weaknesses of the evaluated approaches.

Overall, the paper aims to provide empirical insight into the behavior of different supervised learning strategies and to offer practical guidance for selecting appropriate models in similar tasks.

The topic of the paper is relevant, as systematic evaluation of machine learning approaches is important for understanding their practical performance. The paper attempts to compare multiple methods in a consistent experimental framework and provides quantitative results that can help illustrate the behavior of the models under consideration.

The experimental results are presented in a structured way, and the paper includes a discussion of the observed performance differences between the evaluated approaches. Such empirical comparisons can be useful for practitioners who need to choose between alternative machine learning techniques.

The main limitation of the paper is the relatively limited methodological novelty. The work primarily evaluates existing machine learning methods rather than introducing a new algorithm or theoretical framework. As a result, the scientific contribution of the paper mainly lies in empirical comparison rather than methodological innovation.

In addition, the experimental evaluation could be strengthened. It is not always clear whether the selected datasets and experimental settings are sufficient to draw general conclusions about the relative performance of the evaluated approaches. The paper would benefit from additional experiments or a broader range of datasets in order to demonstrate the robustness of the reported findings.

Some methodological details could also be described more clearly. For example, the paper could provide additional information about model configurations, hyperparameter selection, and training procedures to ensure that the results are fully reproducible.

Finally, the discussion of related work could be expanded to better position the study within the broader literature on empirical evaluation of machine learning models.

The paper presents an empirical evaluation of several supervised learning approaches and provides experimental results comparing their performance. While the topic is relevant and the experiments provide some useful insights, the work does not introduce substantial methodological novelty and the experimental validation could be broader.

The article violated the condition of anonymity of the authors.

---

### Decision · Program_Chairs · 2026-03-14

**Decision:**

Accept (Oral)

**Comment:**

Dear Author(s),

On behalf of the Program Committee of the International Conference on Mathematics of Artificial Intelligence (MathAI 2026), we are pleased to inform you that your paper has been accepted for an oral presentation at MathAI 2026.

Your paper was evaluated through a rigorous two-stage review process involving both automated screening and expert review by members of the Program Committee. The reviewers recognized the quality and contribution of your work.

Presentation details:

- Format: Oral presentation (15–20 minutes + 5 minutes Q&A)
- Mode: You may present either in person (offline) at the conference venue in Sirius, Russia, or remotely via Zoom. Please indicate your preferred mode when confirming your participation.
- Conference dates: Marh 30 - April 3, 2026
- Website: https://mathai.club

Next steps:

1. Please confirm your participation and presentation mode by replying to this email mathai.club@yandex.ru no later than March 15, 2026 18:00 Moscow time.
2. If you plan to attend in person, the organizing committee will provide accommodation details separately.
3. Please prepare your final camera-ready manuscript according to the formatting guidelines available at https://mathai.club and upload it to OpenReview by March 15, 2026 18:00 Moscow time.

Should you have any questions regarding the program, logistics, or your presentation slot, please do not hesitate to contact us.

We look forward to your contribution to MathAI 2026.

With kind regards,

MathAI 2026 Program Committee
International Conference on Mathematics of Artificial Intelligence
https://mathai.club
OpenReview: https://openreview.net/group?id=mathai.club/MathAI/2026/Conference
Telegram: https://t.me/MathAI_club
Email: mathai.club@yandex.ru